# A Wide-Bandwidth PVT-Reconfigurable CMOS Power Amplifier with an Integrated Tunable-Output Impedance Matching Network

**DOI:** 10.3390/mi14030530

**Published:** 2023-02-24

**Authors:** Selvakumar Mariappan, Jagadheswaran Rajendran, Narendra Kumar, Masuri Othman, Arokia Nathan, Andrei Grebennikov, Binboga S. Yarman

**Affiliations:** 1Collaborative Microelectronics Design Excellence Centre (CEDEC), Universiti Sains Malaysia, Bayan Lepas 11900, Malaysia; 2Department of Electrical Engineering, Faculty of Engineering, University of Malaya, Kuala Lumpur 50603, Malaysia; 3Institute of Microengineering and Nanoelectronics, National University of Malaysia, Bangi 43600, Malaysia; 4Darwin College, Cambridge University, Cambridge CB3 9EU, UK; 5Sumitomo Electric Europe Ltd. (UK Office), Hertfordshire WD6 3SL, UK; 6Department of Electrical and Electronics Engineering, Istanbul University, 34320 Istanbul, Turkey

**Keywords:** power amplifier (PA), CMOS, digitally assisted wideband predistorter (DAWPD), back-off output power (PBO), adjacent channel leakage ratio (ACLR), error vector magnitude (EVM), power-added efficiency (PAE), process-voltage-temperature (PVT)

## Abstract

This paper proposes a wideband CMOS power amplifier (PA) with integrated digitally assisted wideband pre-distorter (DAWPD) and a transformer-integrated tunable-output impedance matching network. As a continuation of our previous research, which focused only on linearization tuning for wideband and PVT, this work emphasized improving the maximum output power, gain and PAE across the PVT variations while maintaining the linearity for a wide frequency bandwidth of 1 GHz. The DAWPD is employed at the driver stage to realize a pre-distorting characteristic for wideband linearization. The addition of the tunable-output impedance matching technique in this work provides stable output power, PAE and gain across the PVT variations, through which it improves the design’s robustness, reliability and production yield. Fabricated in CMOS 130 nm with an 8-metal-layer process, the DAWPD-PA with tunable-output impedance matching can achieve an operating frequency bandwidth of 1 GHz from 1.7 to 2.7 GHz. The DAWPD-PA attained a maximum output power of 27 to 28 dBm with a peak PAE of 38.8 to 41.3%. The power gain achieved was 26.9 to 29.7 dB across the targeted frequencies. In addition, when measured with a 20 MHz LTE modulated signal, the DAWPD-PA achieved a linear output power and PAE of 24.0 to 25.1 dBm and 34.5 to 38.8% across the frequency, respectively. On top of that, in this study, the DAWPD-PA is proven to be resilient to process-voltage-temperature (PVT) variations, where it achieves stable performances via the utilization of the proposed tuning mechanisms, mainly contributed by the proposed transformer-integrated tunable-output impedance matching network.

## 1. Introduction

A number of efforts have been made to explore RF integrated circuits (RFICs) for wireless communication applications since global demand is rising rapidly. CMOS technology has been has attracted significant attention in integrated circuit design due to its low cost of implementation and to achieve the goal of realizing complete system-on-chip (SoC) implementation. CMOS-based PAs have numerous limitations because of their low breakdown voltage, the low-quality factor of passive components, high silicon-substrate loss, and the inaccessibility of back via holes for the ground, in which they restrict the attainable linear output power and efficiency as compared to III-V based PAs such as Gallium Arsenide (GaAs) and Gallium Nitride (GaN)-based PAs [1,2,3,4]. Myriad performance-improving methods for linearity and efficiency refinement have been implemented for CMOS PAs to address these aforementioned limitations [5,6].

Efficiency-improvement methods are typically implemented in linear PAs with higher DC-power consumption, while linearity-improvement methods are implemented in switching-based PAs with high non-linearity due to their low conduction angle. Obstacles to achieving linearity in CMOS PAs include the intrinsic non-linear transconductance and non-linear gate-source capacitance C_gs_ [7,8,9]. Thus, it can be deduced that the trade-off between efficiency and linearity in CMOS PAs is inevitable. Along with efficiency and linearity performances, bandwidth improvement in CMOS PAs has been extensively studied [10,11,12]. 

Therefore, a single PA must have a comprehensive frequency bandwidth performance to cater to the countless frequency bands utilized by various wireless communication protocols. This will reduce the number of PAs in a transceiver system and reduce the implementation cost due to the miniaturized chip footprint. However, it is challenging to attain stable linearity and efficiency performances for a wide-bandwidth operation because of the shifting frequency response of the devices. Along with efficiency- and linearity-enhancement techniques, numerous studies on CMOS Pas have emphasized improving the operating bandwidth.

A CMOS dual-mode Doherty PA employing a 4-way symmetrical hybrid transformer is introduced in [13]. An adaptive bias circuit is utilized in the proposed PA to ameliorate its wideband back-off efficiency and linearity. However, at the linear output power region, the efficiency of the PA depreciated by about 10%. Furthermore, a CMOS PA with a transformer-based two-stage dual-radial power splitting and a combining mechanism is proposed in [14]. Some advantages of the transformer include in-phase signal splitting and combining, its compact network, the uniform distribution of DC supplies, and the asymmetric supply and return path of DC. Yet, the PA’s maximum efficiency is lower than other methods because of the splitter’s losses. 

Moreover, a harmonic-traps-based CMOS PA is presented in [15]. The harmonic traps are utilized to reduce the second and third-order harmonics in the PA. The harmonic traps are implemented during the common-source and common-gate stages. The harmonic traps are also implemented in the input matching network, which is realized with a center-tap transformer. Still, its efficiency in linear output power decreased by about 20%. Plus, since the harmonics traps are implemented with inductors, the area consumption on chip is also increased. 

As a continuation of our previous research in [16], which focused only on linearization tuning for wideband and PVT, this work focused on improving the maximum output power, gain and PAE across the PVT while maintaining the linearity for the wide frequency bandwidth of 1 GHz. A reconfigurable digitally assisted wideband pre-distorter (DAWPD) and a tunable-output impedance matching technique were implemented in a three-stage PA which achieves a stable performance throughout the operating frequency and PVT variations. The addition of the tunable-output impedance matching technique in this work provides the stable output power, PAE and gain across the PVT variations, through which it improves the design’s robustness, reliability and production yield. Section 2 explains the architecture and circuit design of the three-stage CMOS DAWPD-PA, digital linearizer and tunable-output impedance matching. Section 3 explains the operation principle of the DAWPD-PA. Section 4 summarises the measurement results of the DAWPD-PA. Finally, Section 5 draws the conclusion.

## 2. Top Architecture and Circuit Details

### 2.1. CMOS DAWPD-PA

The designed three-stage CMOS DAWPD-PA consists of a gain booster, a DAWPD, the main PA, and a tunable-output impedance matching implemented with an interleaved transformer. Figure 1 shows the detailed schematic of the DAWPD-PA. The gain booster is constructed with transistor Q_3_ (1500/0.25 µm). The RC feedback (R_4_ and C_4_) applied in the gain booster improves the stability of the DAWPD-PA. The inductor, L_2_, acts as the RF choke and is supplied with V_DD1_ = 1.2 V. The input matching network is comprised of C_1_, C_2_, C_3_, and L_1_. The DAWPD includes a driver stage, an active load, an interstage matching network, and two digital linearizers (DLs). The DLs reconfigure Q_6_’s and Q_4_’s operation regions. Q_6_ (2500/0.25 µm) is integrated with the active load of Q_4_, Q_5_, C_8_, R_7,_ and R_8_. The DAWPD is supplied with V_DD2_ = 2.5 V. As in Figure 1, the digital linearizer 1 (DL1) is utilized to provide the biasing voltage to the gate of Q_6_ through R_5_. On the other hand, the digital linearizer 2 (DL2) provides the biasing voltage at the gate of Q_4_ through R_8_. Furthermore, the main PA stage (Q_9_) is realized with a size of 5000/0.35 µm and is supplied with V_DD3_ = 3.3 V. The parallel stack LC load consisting of L_5_, C_11_, C_12_, L_6_, C_13_, and C_14_ is utilized to mitigate the drain resistance of the main stage, which further contributes to providing optimum efficiency across the bandwidth.

In addition, to provide an optimum impedance transformation at the output of the main PA, a π-structured matching network consisting of C_15_, L_7_, C_16_, and C_17_ was implemented prior to the tunable-output impedance matching. The π-network matching stage converts the low output impedance of the main PA into a higher impedance value. The employed tunable-output impedance matching with the transformer eliminates the requirement of an additional multistage matching network for wideband operation and PVT reconfiguration. The tunable-output impedance matching offers an impedance tuning characteristic which produces an optimum impedance for a wide frequency bandwidth and PVT variations. Referring to Figure 1, the designed tunable-output impedance matching consists of T_1_, C_18_, C_19_, Q_10_, Q_11_, R_13_, and R_14_. T_1_ is realized with a patterned ground shield (PGS) for the enhancement of the transformer’s Q factor. The impedance at the primary winding side of T_1_ is varied by controlling V_TUNE1_, while the impedance of the secondary winding side of T_1_ is controlled via V_TUNE2_.

### 2.2. Digital Linearizer

As illustrated in Figure 2, the digital linearizer (DL) comprises a seven-stage voltage generator which produces digitally varied coarse and fine voltage tuning. The DL is constructed via the binary-weighted current sources (Y_2_:Y_5_ and Y_20_:Y_22_). Utilizing the current sources, a stable current is supplied to the load resistor (R_O_). The PMOS current sources (Y_2_:Y_5_) are employed to supply a stable current to the resistor (R_O_) which performs coarse voltage tuning using bits A_0_ to A_3_. Meanwhile, the NMOS (Y_20_:Y_22_) is employed to divert the current away from the output node and it performs fine voltage tuning using bits represented by S_0_ to S_2_. The total current going through R_o_ is adjusted by the digital bits [A0:A3] and [S0:S2]. [A0:A3] bits conduct coarse voltage tuning with larger voltage changes for each bit. Bit A3 is the most significant bit (MSB), while bit A0 is the least-significant-bit (LSB).

In addition, [S0:S2] bits conduct fine voltage tuning with small voltage variations between each bit. Bit S2 is the MSB while bit S0 is the LSB. In general, the DL has 128 states of different voltage levels, of which 120 generate usable output voltages which can vary the operating region of the transistors. Moreover, the bandgap reference employed in the digital linearizer offers a stable voltage reference across PVT variations. Referring to Figure 2, by utilizing the bandgap reference voltage, a constant current reference is produced for the seven-stage voltage generator through the op-amp (OA2), Y_7_ and R_7_. OA_2_ is employed in a closed-loop system in order to achieve a constant current reference (I_REF_) for the current mirrors in the seven-stage voltage generator. In addition, unity gain buffers (UGB1 and UGB2) are employed to provide a high impedance path so that a very small current will be drawn from the nodes. UGB2 also provides an isolation from the RF signal in the PA when it is integrated to it. By applying KCL at node X in Figure 2 and taking *R*_O_ generates voltage:(1)$$V_{O}=I_{O}\cdot R_{O}=\left(I_{A}-I_{S}\right)\cdot R_{O}$$
where
(2)$$I_{A}=I_{\mathrm{REF}}\left(A_{0}I_{A0}+A_{1}I_{A1}+A_{2}I_{A2}+A_{3}I_{A3}\right)$$
(3)$$I_{S}=I_{\mathrm{REF}}\left(S_{0}I_{S0}+S_{1}I_{S1}+S_{2}I_{S2}\right)$$

Figure 3 shows the plot of the simulated output voltage levels across the digital bits. 

### 2.3. Tunable-Output Impedance Matching Network Design

The tunability of the output impedance matching network is realized via a tuner circuit consisting of a deep triode common-source transistor, a stabilizer resistor, and a capacitor. The current source tuner (CST) is applied at the transformer in both primary and secondary windings, as depicted in Figure 4. As shown in Figure 4, the primary side of the transformer, L_p_, has a deep triode common source transistor, Q_1,_ which controls its impedance via the tuning voltage (V_TUNE1_). The resistor (R_1_) functions as a stabilizer which prevents Q_1_ from oscillating because of the escalating input RF signal through the transformer. The capacitor, C_1_, acts as the reactance tuner of the transformer. The tuner circuit consists of Q_2_, R_2_, and C_2_ at the secondary winding; L_s_ has the same purpose as previously mentioned. 

As illustrated in Figure 5a, it can be observed that when V_TUNE1_ is varied, the current that flows into node X is controlled, i_1_ flows into the primary winding of the transformer (L_p_) while i_2_ flows into the tuner. The same event occurs following the secondary winding of the transformer (L_s_), when V_TUNE2_ is varied. The current flow is represented by i_3_ and i_4_ at node Y in the secondary winding. Figure 5a,b depicts the simulated current flow at node X and Y, respectively. Figure 6 shows the equivalent circuit of the tunable-output impedance matching network. The impedances of both the primary and secondary sides of the transformer are tuned using the CST.

It can be deduced that the rate of change in current (i_1_ in primary and i_3_ in secondary) into L_p_ and L_s_ are reduced. Therefore, this accords with the increment in inductance values of the transformer windings, as defined by:(4)Lp=VXdi1dt
(5)Ls=VYdi3dt
where *V*_X_ and *V*_Y_ are the voltages at nodes X and Y, respectively. 

Furthermore, based on Figure 6, the parameters from the transformer are Lsn, Rsn, Csn, Coxn, Csin, and Rsin. The external parameters introduced into the transformer are the capacitors (C_1_ and C_2_) and resistors (R_O1_ and R_O2_) from the CST at the primary and secondary windings. The R_O1_ and R_O2_ are the output resistances of Q_1_ and Q_2_, respectively. The admittance following the primary and secondary windings of the transformer are expressed in terms of Y_11_ and Y_33_, respectively [17,18,19]. When simulating or measuring the impedance of the transformer, the parameters of the primary winding are obtained by keeping the secondary winding open [20]. The inverse is performed when obtaining the parameters of the secondary winding. From Figure 6, Y_11_ and Y_33_ are derived and expressed as follows:(6)$$\begin{array}{ll} Y_{11} & =\frac{1}{R_{s1}+j\omega L_{s1}}+j\omega C_{s1}\\ & +\left[j\omega C_{\mathrm{ox}1}\left\|\left(\frac{1}{R_{\mathrm{si}1}}+j\omega C_{\mathrm{si}1}\right)\right.\right]+\left[j\omega C_{1}\left\|\frac{1}{R_{O1}}\right.\right] \end{array}$$
(7)$$\begin{array}{ll} Y_{33} & =\frac{1}{R_{s2}+j\omega L_{s2}}+j\omega C_{s2}\\ & +\left[j\omega C_{\mathrm{ox}3}\left\|\left(\frac{1}{R_{\mathrm{si}3}}+j\omega C_{\mathrm{si}3}\right)\right.\right]+\left[j\omega C_{2}\left\|\frac{1}{R_{O2}}\right.\right] \end{array}$$
where *R*_O1_ and *R*_O2_ are expressed as:(8)$$R_{O1}=\frac{1}{i_{2}\lambda_{1}}=\frac{1}{\left(\frac{1}{2}\mu_{n1}C_{\mathrm{ox}1}\frac{W_{1}}{L_{1}}{\left(V_{\mathrm{TUNE}1}-V_{\mathrm{TH}1}\right)}^{2}\right)\lambda_{1}}$$
(9)$$R_{O2}=\frac{1}{i_{4}\lambda_{2}}=\frac{1}{\left(\frac{1}{2}\mu_{n2}C_{\mathrm{ox}2}\frac{W_{2}}{L_{2}}{\left(V_{\mathrm{TUNE}2}-V_{\mathrm{TH}2}\right)}^{2}\right)\lambda_{2}}$$
where *μ*_n*n*_ is the electron mobility, *C*_ox*n*_ is the oxide capacitance, *W_n_* is the width of the transistor, *L_n_* is the length of the transistors, V_TUNE*n*_ is the gate voltage, V_TH*n*_ is the threshold voltage, and λ*_n_* is the channel length modulation of the transistors. The subscripted *n* represents the transistors (Q_1_ and Q_2_) from the CST.

It can be deduced from (8) and (9) that V_TUNE*n*_ is inversely proportional to R_O*n*_, where it contributes to the impedance tunability. The simulated impedance values of the tunable-output impedance matching network are depicted in Figure 7. It can be observed that the impedance locations on the Smith chart vary when the CST is tuned for different frequencies. The variation in the impedance value is helpful in realizing a tunable impedance matching network for the PA.

## 3. Operation Principle

### 3.1. DAWPD Mechanism

In a PA, the transconductance (g_m_) of the transistor, especially the third-order transconductance (g_m3_) is one of the primary contributors to distortions. The third-order distortions occurring at the frequencies of 2ω_1_-ω_2_ and 2ω_2_-ω_1_ are highly focused because they appear in-band and are extremely difficult to filter out. The transfer function of a PA can be derived using the Taylor series expansion, which is given by [21]:(10)$$\begin{array}{ll} i_{\mathrm{out}}\left(v_{\mathrm{in}}\left(t\right)\right) & =\frac{dI_{\mathrm{DS}}}{dV_{{}_{\mathrm{GS}}}^{}}*v_{\mathrm{in}}\left(t\right)+\frac{1}{2!}\frac{d^{2}I_{\mathrm{DS}}}{dV_{{}_{\mathrm{GS}}}^{2}}*v_{\mathrm{in}}^{2}\left(t\right)+\frac{1}{3!}\frac{d^{3}I_{\mathrm{DS}}}{dV_{{}_{\mathrm{GS}}}^{3}}*v_{\mathrm{in}}^{3}\left(t\right)+\ldots \\ & =g_{m1}*v_{\mathrm{in}}\left(t\right)+g_{m2}*v_{\mathrm{in}}^{2}\left(t\right)+g_{m3}*v_{\mathrm{in}}^{3}\left(t\right)+\ldots \end{array}$$
where *v_in_(t)* is the input voltage, *g_mn_* is the n^th^-order coefficient of the transconductance and *i_out_* is the output current. As seen in (10), the g_m3_ is the third-order transconductance coefficient contributing to IMD3 generation. 

To delineate the cancellation operation of the IMD3 products in the spectrum domain, a two-tone test can be utilized in which the input signal is represented by:(11)$$v_{\mathrm{in}}\left(t\right)=A\left[\cos\left(\omega_{1}t\right)+\cos\left(\omega_{2}t\right)\right]$$

By assuming the upper and lower IMD signals are equal and considering only the upper IMD3 components of the DAWPD and main PA at 2ω_2_ - ω_1_, the I_out_ (*v*_in_) is given as [22]:(12)$$i_{\mathrm{out},\mathrm{dawpd}}\left(2\omega_{2}-\omega_{1}\right)=\frac{3A^{3}}{4}g_{m3,\mathrm{dawpd}}e^{j\theta_{\mathrm{dawpd}}}$$
(13)$$i_{\mathrm{out},\mathrm{main}}\left(2\omega_{2}-\omega_{1}\right)=\frac{3A^{3}}{4}g_{m3,\mathrm{main}}e^{j\theta_{\mathrm{main}}}$$
where *θ*_main_ and *θ*_dawpd_ are the phases of the IMD3 product, while g_m3,main_ and g_m3,dawpd_ are the third-order transconductance of the main and DAWPD. Referring to (12) and (13), it can be concluded that to minimize the IMD3 distortions, the g_m3_ and θ of both the DAWPD and main PA stages should be in contrast to each other, as in (14):(14)$$g_{m6,3}Z_{\mathrm{dawpd},3}=-g_{m9,3}Z_{\mathrm{main},3}$$

Based on (14), it is deduced that to attain the ideal IMD3 cancellation, the third-order components produced at the DAWPD’s output need to have a 180° out-of-phase response compared to the third-order components produced at the main PA’s output [23]. The opposite characteristic between the stages can be perceived in terms of opposite AM-AM and AM-PM behavior based on the fundamental components. Referring to Figure 8, the g_m3_ of the DAWPD is varied utilizing the DL employed at the gate of the Q_6_. The transconductances of the DAWPD and main PA are determined, and optimum operating conditions are selected. From (14), it can be seen that to conduct the IMD3 cancellation, the g_m6,3_ should be in contrast to g_m9,3_. Thus, the selected operating region for the DAWPD and main PA is shown in Figure 8. Herein, it can be seen that the g_m3_ of both DAWPD and main PA contrast with each other. The main PA’s g_m3_ is −7.3 A/V^3^ at V_GS_ = 0.8 V. On the other hand, the DAWPD’s g_m3_ was selected to be +7.3 A/V^3^ at V_GS_ = 0.42 V or V_GS_ = 0.58 V, since both of these V_GS_ values can be utilized to provide the cancellation. However, since the higher V_GS_ provides a higher gain, the V_GS_ = 0.58 V was selected here.

The V_GS_ of the DAWPD was selected via the digital bits combination from the digital linearizer. Here, the A-bits are represented by the first four bits, while the S-bits are represented by the last three bits. For example, the V_GS_ = 0.58 V was selected by tuning the A-bits to 0100 and S-bits to 010, which gives a full combination of 0100010. It can be observed that the DAWPD can be operated at two different regions which generate equal opposite g_m3_ to that of the main PA. The operating region of the DAWPD can be varied via the digital linearizer to achieve optimum performance across the wideband operation and PVT variations. Furthermore, the small-signal equivalent circuit of the DAWPD and main PA is depicted in Figure 9a,b, respectively. The output impedances of DAWPD and the main PA stages are derived based on the equivalent circuit. The output impedances of the DAWPD (Z_dawpd_) and main PA (Z_main_) stages are shown in (15) and (16), respectively. The output impedance of the DAWPD is inclusive of C_gs9_ from the main stage as its load, since it is in a cascade with the main stage. The C_gs9_ is also a part of the fully capacitive response in the DAWPD.
(15)$$Z_{\begin{array}{l} \mathrm{dawpd} \end{array}}=\left[\left[\left[\left[\left(R_{o6}\left\|R_{o4}\right.\right)\left\|\left(R_{o5}+R_{6}-\frac{j}{\omega C_{8}}\right)\right.\right]-\frac{j}{\omega C_{9}}\right]\left\|-\frac{j}{\omega C_{10}}\right.\right]+j\omega L_{4}+R_{10}\right]\left\|-\frac{j}{\omega C_{\mathrm{gs}9}}\right.$$
(16)$$Z_{\mathrm{main}}=\frac{j\omega L_{5}L_{6}R_{o9}}{((1-\omega^{2}\left(C_{11}+C_{14}\right)L6)L_{5}+L_{6})R_{o9}+j\omega L_{5}L_{6}}$$

As observed in (15), a capacitive response influences DAWPD’s output impedance. Meanwhile, the main PA’s output impedance is dominated by an inductive response which is contributed by the large L_5_ and L_6_ values from the designed parallel LC stack. Both stages’ exhibited capacitive and inductive responses provide the required opposite-phase characteristics. The interstage impedance between the DAWPD and main PA is vital since it reflects the characteristic of the pre-distorted signal, which is fed into the main PA’s input in which it impacts the linearity of the PA across the wide operating bandwidth [24,25]. Hence, the interstage impedance is beneficial in attaining a wideband linearization. Since the operating region of Q_4_ in the active load is tuned, it provides a variable impedance at the DAWPD’s output (Z_dawpd_). The tuning of Q_4_ varies the output resistance which R_o4_ represents, as in (15). Based on (15), R_o4_ and R_o6,_ which are dependent on the tuning of the digital linearizer, appear as part of the DAWPD’s output impedance, contributing to the configurability of the impedance which is beneficial for wideband linearity tuning. 

The opposite response of the PA is simulated across the operating bandwidth from 1.7 to 2.7 GHz. The results are at 1.7, 2.45, and 2.7 GHz, where the PA’s performance is optimum. The opposite AM-AM response is depicted in Figure 10a. Since the main PA’s AM-AM profiles expand across output power at different operating frequencies, the DAWPD is tuned to exhibit a contrary compressed AM-AM profile. As seen in Figure 10a, the digital bits variations in the DAWPD generate the optimum compressing responses needed to cancel out the expanding gain of the main PA.

The resultant flat gain AM-AM responses across the output power at different frequencies are depicted in Figure 10b. At 1.7 GHz, the resultant AM-AM is flat across the output power with a deviation by ±0.2 dB up to the output power of 27 dBm. At 2.45 and 2.7 GHz, the AM-AM responses are flat with a deviation by ±0.2 dB up to the output power of 28.6 and 28.5 dBm, respectively. This illustrates the effectiveness of the DAWPD tuning in enhancing the AM-AM performance of the PA across the wideband frequency. 

Moreover, the AM-PM responses are also analyzed across the frequencies mentioned above. The DAWPD can produce a phase response opposite to the phase response of the main PA. The opposite-phase responses between the stages are compensated by each other, through which a flat phase response across output power is achieved. The opposite phase responses and their resultant AM-PM are illustrated in Figure 11a,b, respectively. Figure 11b shows that the resultant AM-PM deviation is about 4° up to the output power of 26 dBm at 1.7 GHz. Deviations by 4° in AM-PM are achieved up to 27.6 and 27.3 dBm at 2.45 and 2.7 GHz, respectively. An AM-PM deviation by 5° or more is approximately equivalent to a gain compression of 1 dB or more [26]. The AM-PM flatness achieved indicates the linearization mechanism using DAWPD is effective. 

The DAWPD’s effectiveness on wideband linearization is also further validated through the simulation of IMD3. The IMD3 simulation was carried out across the operating bandwidth and is presented before and after DAWPD integration in Figure 12. By adopting the IMD3 specification of −30 dBc, it is observed that with DAWPD’s integration, the PA can meet the specification up to 25.5 dBm output power at 2.45 GHz. Meanwhile, at 1.7 and 2.7 GHz, the PA can meet the specification up to 24 and 23.5 dBm, respectively. After integrating and tuning the DAWPD, it is observed that the IMD3 was significantly enhanced at high output power levels. The IMD3 is extended up to 28 dBm at 2.45 GHz, 27.2 dBm at 1.7 GHz, and 27 dBm at 2.7 GHz. The digital bits of the DAWPD were tuned to achieve the low peaks of IMD3, which occur from −50 to −60 dBc at high output-power levels across the frequency. The low peaks of the IMD3 are regarded as the “sweet spot,” which reflects the region of the maximum IMD3 product cancellation. 

### 3.2. Tunable-Output Impedance Matching Network

As aforementioned, the main PA stage was designed with a size of 5 mm to achieve a maximum output power of 30 dBm. However, the PA delivers the maximum output power at its lowest output impedance, as illustrated in the load-pull analysis in Figure 13. The load-pull analysis was conducted at 2.45 GHz. Referring to the load-pull analysis, the PA delivers the maximum output power of 30 dBm at an impedance of 3 Ω. Hence, the output impedance of the main PA needs to be precisely matched to the output load impedance of 50 Ω for maximum power transfer. It is complicated to design a simple impedance transformation matching network which transforms directly from 3 to 50 Ω. This commonly requires multiple stages of a matching network which gradually transform the impedance, especially for wideband applications. The multistage matching network increases the area consumption on-chip and its implementation cost. Thus, the proposed tunable-output impedance matching network eradicates the need for higher order multistage matching networks. The π network shifts the low output impedance (3 Ω) of the main PA to a higher impedance (22 Ω) so that the tunable-output impedance matching network can be designed with low complexity to further shift the output impedance close to the port impedance (50 Ω). The locations of the impedance points are shown in a block diagram as in Figure 14, and its impedance transformation process is illustrated using the Smith chart in Figure 15.

The impedance transformation for maximum output power at 2.45 GHz is taken as an example for delineation. Based on the aforementioned figures, the impedance location at the drain of the PA is represented by Z_main,_ and the impedance location after the π-network is shown by Z_π-net_. Meanwhile, the impedance location after implementing the tunable-output impedance matching network is represented by Z_L_. Overall, the output matching network shifts the location of the impedance from Z_main_ to Z_L_. 

The wideband operation and the PVT robustness of the DAWPD-PA is realized via the tunable-output impedance matching network in which the output impedance, Z_L_, across the frequencies is shifted near to 50 Ω by varying the V_TUNE1_ and V_TUNE2._ This gives tunability of the impedance at the primary and secondary sides of the transformer. As delineated in Figure 16, it can be observed that the output matching network can only provide an impedance near to 50 Ω for limited frequencies. Based on Figure 16, it is also observed that the impedance location for 1.7, 2.45, and 2.7 GHz can be brought near to 50 Ω with tuning, which reflects the effectiveness of the tunable-output impedance matching network at tuning the output impedance of the DAWPD-PA.

The Z_L_ is derived based on the equivalent circuit depicted in Figure 17. Referring to Figure 17, the Z_main_ is the main PA’s output impedance, expressed in (16). C_15_, L_7_, C_16,_ and C_17_ are the π network implemented before the tunable-output impedance matching network. C_18_ and R_O10_ are the primary CST components, in which R_O10_ is the output resistance of Q_10_ while L_p_ is the inductance of the primary winding. On the secondary side, C_19_ and R_O11_ are the secondary CST components in which R_O11_ is the output resistance of Q_11_ while L_s_ is the inductance of the secondary winding. The *k* is the coupling factor of the transformer. To simplify the analysis, the transformer in the circuit is replaced with a T-model equivalent circuit, and the schematic is modified, as illustrated in Figure 18 [27]. 

Referring to Figure 18, the transformer is now represented by the inductors (L_p_-M), (L_s_-M), and M, in which M is the mutual inductance between the primary and secondary windings and can be derived as [28]:(17)$$M=k\sqrt{L_{p}L_{s}}$$

For simplicity of analysis, the Z_L_ is derived stage by stage. The impedance after the implementation of the π network, Z_π-net_ is derived as: (18)$$Z_{\pi -\mathrm{net}}=\frac{-j\omega^{2}\left(L_{7}Z_{\mathrm{main}}\left(C_{15}+C_{16}+C_{17}\right)\right)-jZ_{\mathrm{main}}-\omega L_{7}}{\omega^{3}\left(L_{7}C_{17}Z_{\mathrm{main}}\left(C_{15}+C_{16}\right)\right)-j\omega^{2}\left(L_{7}C_{17}\right)-\omega C_{17}Z_{\mathrm{main}}}$$

Hence, Z_L_ is derived as in (19). Referring to (19), it can be perceived that the CST components appear in both the real and imaginary parts of the equation, in which they provide the variation in the magnitude and phase of the Z_L_. When the CST is turned off, the effect of C_18_, C_19_, R_O10_, and R_O11_ will not vary Z_L_ and, thus, can be eliminated from (19). Therefore, Z_L_ when the CST is turned off is simplified as derived in (20).


(19)
$$Z_{L}=\frac{\begin{array}{c} \omega^{4}\left[\left(L_{p}L_{{}_{s}}R_{O10}+L_{p}L_{{}_{s}}Z_{\pi -\mathrm{net}}-M^{2}R_{O10}-M^{2}Z_{\pi -\mathrm{net}}\right)C_{18}C_{19}R_{O11}\right]\\ -j\omega^{3}\left[\left(\left(\left(C_{19}R_{O11}Z_{\pi -\mathrm{net}}+L_{p}\right)R_{O10}+L_{p}Z_{\pi -\mathrm{net}}\right)L_{{}_{s}}-M^{2}\left(R_{O10}+Z_{\pi -\mathrm{net}}\right)\right)C_{18}-C_{19}R_{O11}\left(M^{2}-L_{p}L_{s}\right)\right]\\ -\omega^{2}\left[\left(\left(C_{18}R_{O10}+C_{19}R_{O11}\right)Z_{\pi -\mathrm{net}}+L_{p}\right)L_{s}-M^{2}\right]+j\omega L_{s}Z_{\pi -\mathrm{net}} \end{array}}{\begin{array}{c} \omega^{4}\left[\left(L_{p}L_{{}_{s}}R_{O10}+L_{p}L_{{}_{s}}Z_{\pi -\mathrm{net}}-M^{2}R_{O10}-M^{2}Z_{\pi -\mathrm{net}}\right)C_{18}C_{19}\right]\\ -j\omega^{3}\left[\left(\left(L_{p}R_{O11}\left(R_{O10}+Z_{\pi -\mathrm{net}}\right)+L_{s}Z_{\pi -\mathrm{net}}R_{O10}\right)C_{18}+L_{p}L_{{}_{s}}-M^{2}\right)C_{19}\right]\\ -\omega^{2}\left[\left(C_{19}Z_{\pi -\mathrm{net}}R_{O10}R_{O11}+L_{p}\left(Z_{\pi -\mathrm{net}}+R_{O10}\right)\right)C_{18}+\left(R_{O11}L_{p}+L_{{}_{s}}Z_{\pi -\mathrm{net}}\right)C_{19}\right]\\ -j\omega \left[\left(C_{18}R_{O10}+C_{19}R_{O11}\right)Z_{\pi -\mathrm{net}}+L_{p}\right]+Z_{\pi -\mathrm{net}} \end{array}}$$



(20)
$$Z_{L_{\mathrm{CST}:\mathrm{OFF}}}=\frac{\omega^{2}\left(M^{2}-L_{p}L_{s}\right)+j\omega \left(L_{p}Z_{\pi -\mathrm{net}}\right)}{Z_{\pi -\mathrm{net}}+j\omega L_{p}}$$


Figure 19a shows the simulated S parameters of the DAWPD-PA with the tunable-output impedance matching network. The simulated performances were obtained from the optimum settings of both the digital bits of the DAWPD and the tunable-output impedance matching network. It can be observed that the resonance points of S_11_ and S_22_ shifted across the frequency by tuning the tunable-output impedance matching network. The S_11_ and S_22_ are made to achieve −10 dB across the operating bandwidth to provide optimum input and output return losses, respectively. Referring to Figure 19a, it can be seen that the bandwidth is limited to cover the operating frequency of 1.7 to 2.7 GHz with a single setting. Therefore, the tunable-output impedance matching network is employed to achieve an optimum small-signal parameter across the operating frequency bandwidth. At 1.7 GHz, the S_11_ achieved is −23 dB while the S_22_ is −11 dB. At 2.45 GHz, the S_11_ is −25 dB, while the S_22_ achieved is −23 dB. Last but not least, at 2.7 GHz, the S_11_ is −15 dB while the S_22_ is −28 dB. The small-signal gain (S_21_) performances do not significantly vary across the frequencies in which it achieved a gain of 30, 31, and 28 dB at 1.7, 2.45, and 2.7 GHz, respectively. Different settings of DAWPD and the tunable-output impedance matching network are utilized to achieve optimum performances across the frequency.

Furthermore, the DAWPD-PA with the tunable-output impedance matching network delivers a maximum output power of 30.2 dBm when simulated at 2.45 GHz. Prior to tuning, the maximum output power degrades by about 1 dB when simulated at 1.7 and 2.7 GHz. After tuning the integrated tunable-output impedance matching network, the maximum output power is enhanced across the frequency bandwidth. The drop in maximum output power is reduced to about 0.5 dB across the frequency. The simulated output power at 1.7 and 2.7 GHz are enhanced to 29.9 and 29.7 dBm, respectively. This validates the effectiveness of the tunable-output impedance matching network in providing an optimum impedance across the bandwidth for high output power and efficiency. Figure 19b shows the simulated power gain and output power performances across the operating bandwidth. It can be observed that the gain and maximum output power are slightly enhanced across the frequency.

The PAE performance of the DAWPD-PA was also simulated prior and after tuning the tunable-output impedance matching network as depicted in Figure 19c. It is noticeable that prior to tuning, the PAE significantly dropped, by about 8%, when the operating frequency is shifted to 1.7 and 2.7 GHz. Prior to tuning, the peak PAE achieved was 45% at 2.45 GHz, 37% at 1.7 GHz, and 38% at 2.7 GHz. However, the PAE was significantly enhanced across the frequency after tuning the tunable-output impedance matching network. At 2.45 GHz, the peak PAE achieved is 46% while 44% was achieved at both 1.7 and 2.7 GHz. The PAE was enhanced by 7% compared to the PAE prior to tuning, at 1.7 and 2.7 GHz.

In addition, the DAWPD-PA was also simulated across the PVT variations at 2.45 GHz in order to validate its robustness and reliability as compared to our previous research in [16]. The DAWPD-PA was simulated at different corners such as the TT (typical–typical), SS (slow–slow), and FF (fast–fast) corners as well as at different temperatures of +27°, −25°, and +125°. The PVT performances are shown prior to and after tuning the DAWPD-PA in order to validate its reconfigurable feature under PVT variations which degrade its performance. The PAE and power-gain performances at different corners are shown in Figure 20a,b, respectively. From Figure 20a, it can be seen that the peak PAE at FF is better by 4% than its TT value of 46%. For SS, the peak PAE degrades by 8% prior to tuning, which yields a peak PAE of 38%. After tuning both the DAWPD and tunable transformer, the peak PAE is enhanced to 42%, which is an increment by 4% from the lower value. Referring to Figure 20b, the power gain shows a similar trend, wherein it is better in FF and degrades in SS. Prior to tuning, the power gain degrades from 31.5 dB to 27.5 dB in SS and it is enhanced to 29.5 dB after tuning.

Furthermore, the PAE and power-gain performances at different temperatures are illustrated in Figure 21a,b, respectively. From Figure 21a, it can be seen that the peak PAE is higher at −25°, by 6%, than its +27° value of 46%. Prior to tuning, the peak PAE degrades by 9% at +125°, which results in a peak PAE of 37%. After tuning, it is feasible to improve the peak PAE to 40%, which is an increment of 3% over its degraded value. Meanwhile, the power gain is also increased at −25° and deteriorated at +125°. Prior to tuning at +125°, the power gain degrades from 31.5 dB to 25.5 dB and it is enhanced to 28.5 dB after tuning, as shown in Figure 21b. The configurability of the DAWPD-PA with tunable-output impedance matching network is validated via the PVT simulation in which the tuning of the digital bits in the DAWPD and tuning of the output impedance matching network improve its performances. Hence, the proposed techniques enhance the functionality of the PA when it is affected by PVT variations as well as contributing to wide-bandwidth operation.

## 4. Measurement Results

Figure 22 shows the micrograph of the fabricated DAWPD-PA. From our previous study in [16], which included a fixed-output impedance matching network, the design has been modified by adding the CST to the transformer output matching network in order to realize the tunability mechanism. The added tuner is varied via the bond pad employed for V_TUNE1_ and V_TUNE2_. Fabricated in 130 nm CMOS technology with eight metal layers, the DAWPD-PA consumes an area of 2.37 mm^2^ on-chip, including the bond pads for measurement. The DAWPD-PA was first measured for small-signal performance with a continuous-wave (CW) signal, and Figure 23a shows the S parameters when the digital states are varied for the operating frequency of 1.7, 2.45, and 2.7 GHz. The measured S_11_ were −17.8, −16.4, and −12.1 dB at 1.7, 2.45, and 2.7 GHz. Meanwhile, the measured S_22_ were −13.4, −16.2, and −19.7 dB at 1.7, 2.45, and 2.7 GHz, respectively. In addition, the S_21_ attained were 28.2 dB at 1.7 GHz, 29.1 dB at 2.45 GHz, and 26.7 dB at 2.7 GHz. Furthermore, the DAWPD-PA was also measured for large-signal performances with the CW signal. Figure 23b shows the power-gain performance, and it can be seen that the attained power gains across frequencies were 28.9, 29.7, and 26.9 dB at 1.7, 2.45, and 2.7 GHz, respectively. The saturated output powers delivered across the frequencies were 27.8 dBm at 1.7 GHz, 28.1 dBm at 2.45 GHz, and 27.2 dBm at 2.7 GHz.

The AM-PM performance of the DAWPD-PA was also measured, and its phase deviation versus output power at different frequencies with its respective tuning states are illustrated in Figure 24a. The measured phase deviations of 4° were observed up to an output power of 24.9 dBm at 1.7 GHz, 25.4 dBm at 2.45 GHz, and 24.3 at 2.7 GHz. A phase deviation of 5° or more indicates the 1-dB compression point where the PA enters the non-linear region. In addition, the PAE performance across the frequencies is depicted in Figure 24b. The peak PAE attained at 1.7 GHz was 38.8%, while at 2.45 and 2.7 GHz, the peak PAE was 41.3% and 38.9%, respectively. In addition, the PAE obtained at 3-dB backed-off output power for the frequencies of 1.7, 2.45, and 2.7 GHz are 34.5%, 38.8%, and 36.3%, respectively. The 3-dB P_bo_ is the linear operating region where the DAWPD-PA satisfies the adjacent-channel-leakage-ratio (ACLR) specification when tested with a 20 MHz/16 QAM LTE-modulated signal.

Figure 25a shows the ACLR attained for the different frequencies, and it can be observed that the ACLR specification of −30 dBc was fulfilled up to an output power of 24.3 dBm at 1.7 GHz, 25.1 dBm at 2.45 GHz, and 24 dBm at 2.7 GHz. In addition, the error vector magnitude (EVM) performance of the DAWPD-PA was also measured, and its result is depicted in Figure 25b. The preferred EVM specification is less than −27.96 dB (4%), indicating the optimum in-band linearity requirement when tested with a 16 QAM LTE signal. Referring to Figure 25b, it can be observed that at 2.45 GHz, the EVM achieved is less than −27.96 dB (4%) up to an output power of 25.7 dBm. At 1.7 and 2.7 GHz, the EVM is within specification up to output powers of 25.5 and 24.3 dBm, respectively.

The measured ACLR and EVM of the DAWPD-PA with tuned conditions are also illustrated via the power spectral density (PSD) and EVM constellation diagram. The PSD measured at different frequencies is depicted in Figure 26a–c, where the spectral mask of the modulated signal is shown. The sidebands are kept within the specification of −30 dBc up to their linear output power across the frequencies (24 dBm at 1.7 and 2.7 GHz, 25 dBm at 2.45 GHz). On the other hand, it can be seen from the measured EVM constellation diagram that the symbols fall within the constellation with minimum deviation when measured at maximum linear output power. Figure 27a–c illustrates the EVM constellation diagram at the aforementioned frequencies.

On top of that, 10 chip samples or device under tests (DUTs) were tested to validate the resilience and reliability of the DAWPD-PA when subjected to PVT variations. The performance consistency of the designed DAWPD-PA is justified by measuring different dice from the wafer at different temperatures and corners. The measurement was conducted at 2.45 GHz for all the 10 DUTs. The tunable-output impedance matching network and the digital linearizer were adjusted to attain the optimum performances when the PA is affected by the PVT variations. The DAWPD-PA’s reliability and manufacturing yield increase because most of the fabricated designs are usable and functioning as intended. Figure 28 shows the linear PAE, linear output power gain, and ACLR performances at 2.45 GHz across the PVT variations for 10 samples. Table 1 delineates the performance summary of the DAWPD-PA and comparison with other recent CMOS PAs.

## 5. Conclusions

The 130 nm wideband CMOS DAWPD-PA with tunable-output impedance matching network delivered an operating bandwidth from 1.7 to 2.7 GHz. The DAWPD with a digital implementation and active load offers a tunable wideband linearization across the aforementioned frequency. It also utilizes a tunable impedance matching network realized via a transformer and tuner circuits to reconfigure its performances when impacted by PVT variations. The maximum output power attained by the wideband DAWPD-PA is 27 to 28 dBm with a peak PAE of 38.8 to 41.3%. A linear output power of 24 to 25.1 dBm with linear PAE of 34.5 to 38.8% was achieved when tested with a 20 MHz LTE-modulated signal. The addition of the tunable-output impedance matching technique in this work provides stable output power, PAE and gain across the PVT variations in which it improves the design’s robustness, reliability and production yield. Its performances can be maintained via the design’s tuning property, which is proven when 10 DUTs were tested across different corners and temperatures. The tuning of the digital-bit states of the DAWPD and the tuning of the tunable-output impedance matching network realize the reconfigurable wideband CMOS PA with high robustness.

## Figures and Tables

**Figure 1 micromachines-14-00530-f001:**
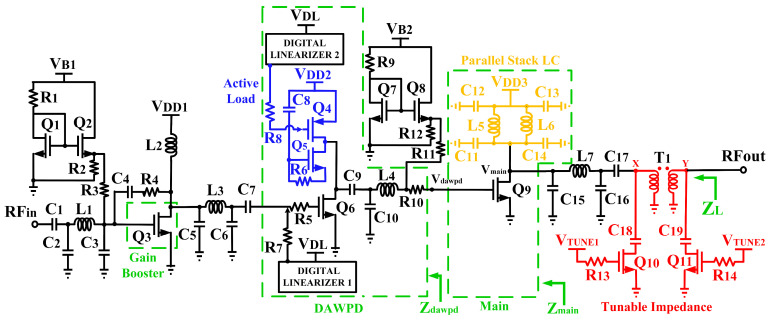
The schematic of the DAWPD-PA with integrated digital linearizer and tunable-output impedance matching network.

**Figure 2 micromachines-14-00530-f002:**
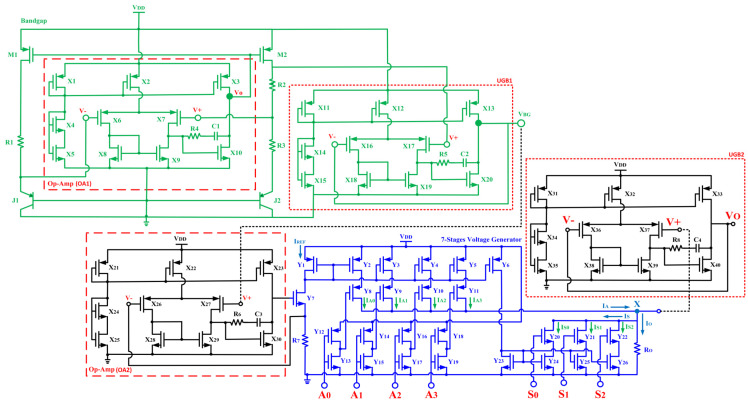
The complete schematic of the integrated digital linearizer.

**Figure 3 micromachines-14-00530-f003:**
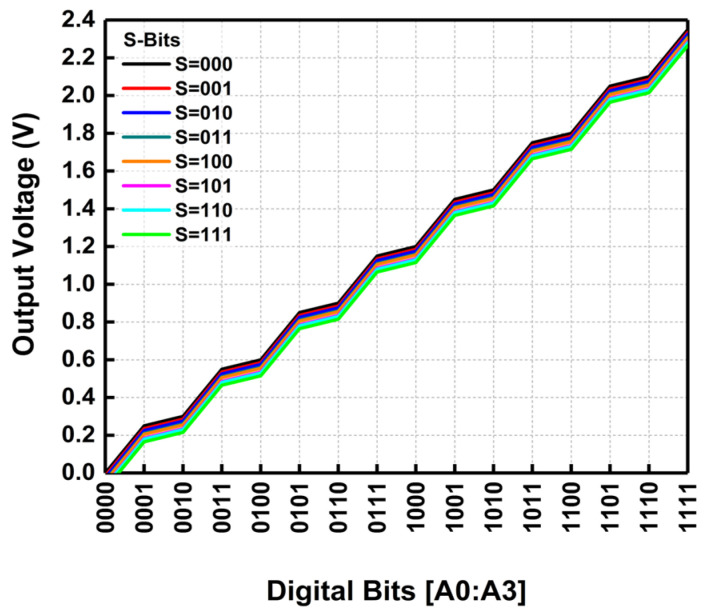
The simulated output voltage generated across the digital bits.

**Figure 4 micromachines-14-00530-f004:**
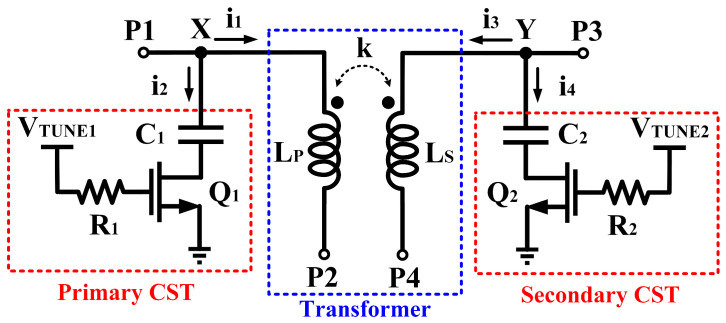
Schematic of the tunable-output impedance matching network with integrated CST.

**Figure 5 micromachines-14-00530-f005:**
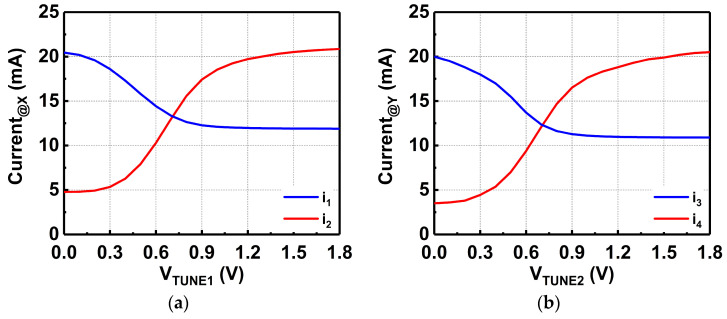
Simulated current at node (**a**) X and node (**b**) Y.

**Figure 6 micromachines-14-00530-f006:**
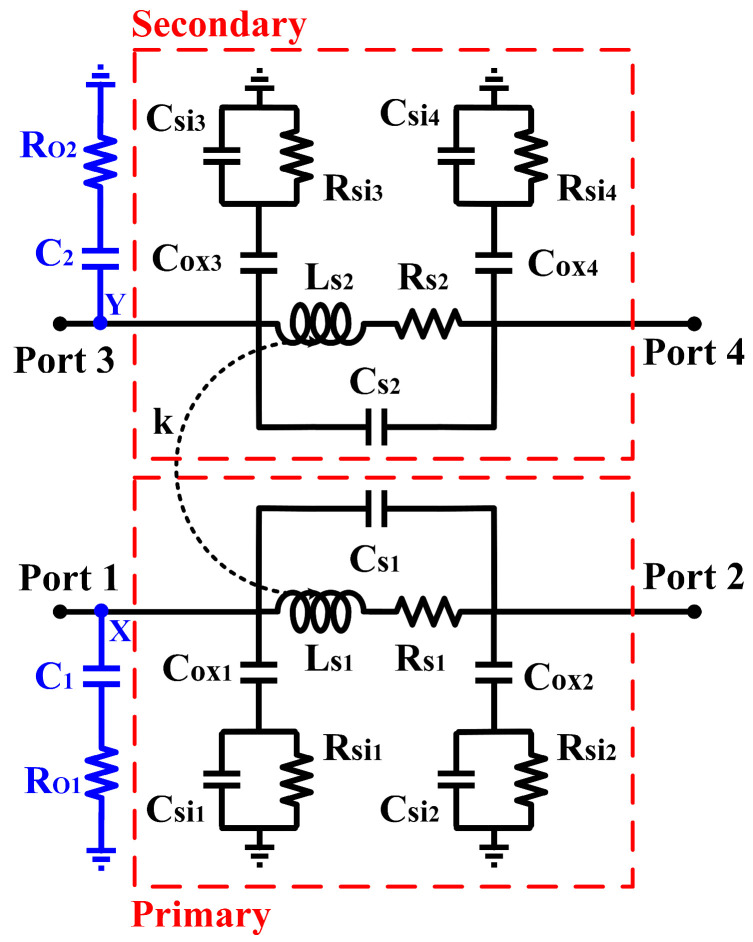
The equivalent circuit of the tunable-output impedance matching network with integrated CST.

**Figure 7 micromachines-14-00530-f007:**
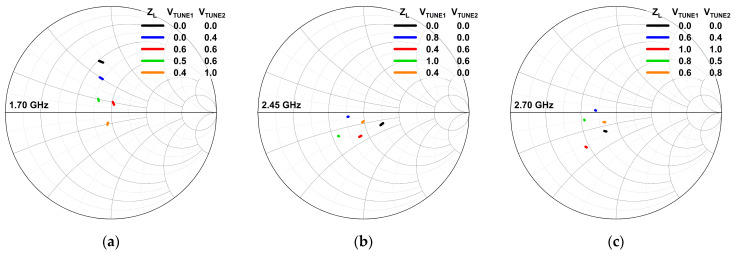
Impedance variation in the tunable impedance matching network at (**a**) 1.7 GHz, (**b**) 2.45 GHz and (**c**) 2.7 GHz.

**Figure 8 micromachines-14-00530-f008:**
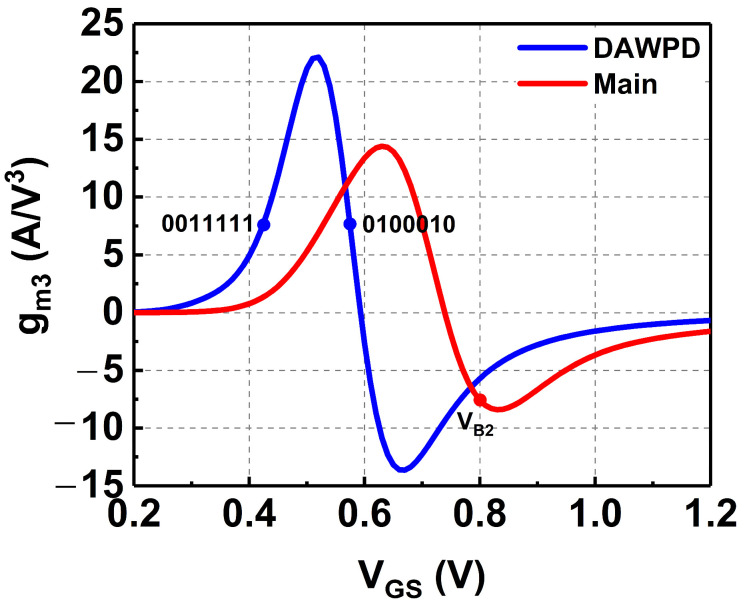
The simulated g_m3_ of the DAWPD and main PA.

**Figure 9 micromachines-14-00530-f009:**
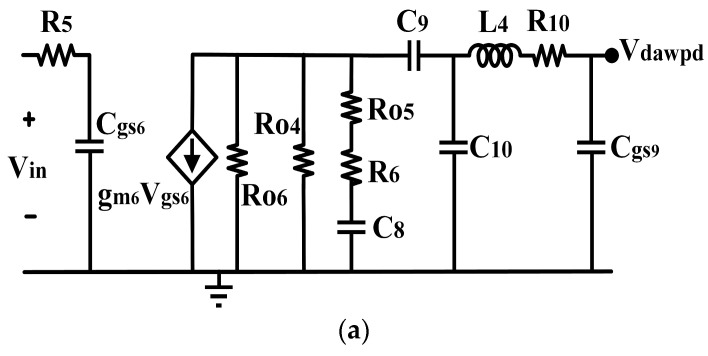
Small-signal equivalent circuit of the (**a**) DAWPD and (**b**) the main PA.

**Figure 10 micromachines-14-00530-f010:**
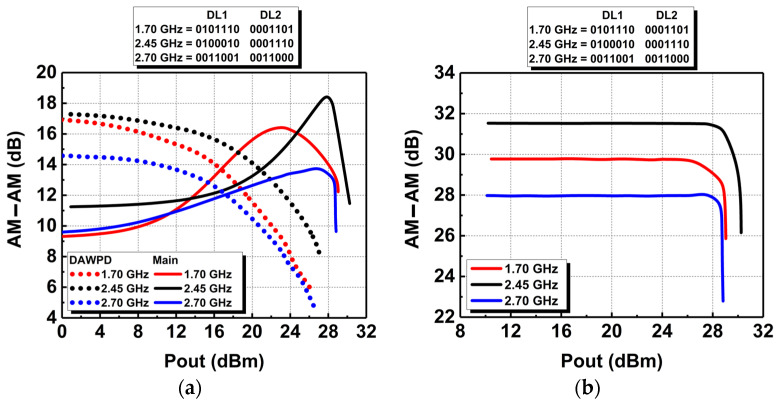
(**a**) Opposite AM-AM response of the DAWPD and the main PA and (**b**) the resultant flat AM-AM response of the integrated PA.

**Figure 11 micromachines-14-00530-f011:**
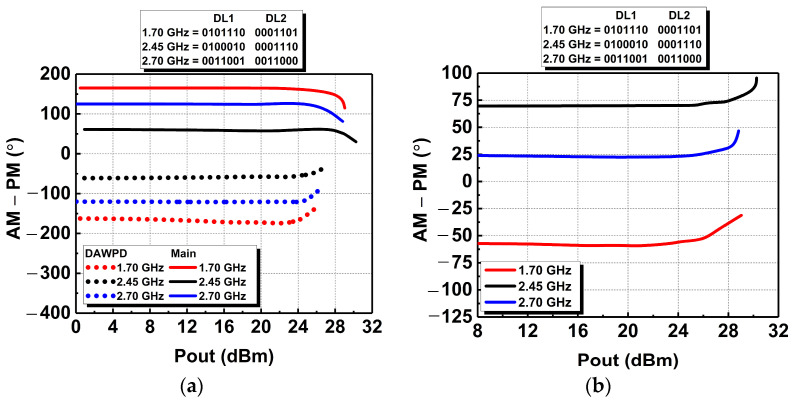
Simulated AM-PM response (**a**) prior and (**b**) after integration.

**Figure 12 micromachines-14-00530-f012:**
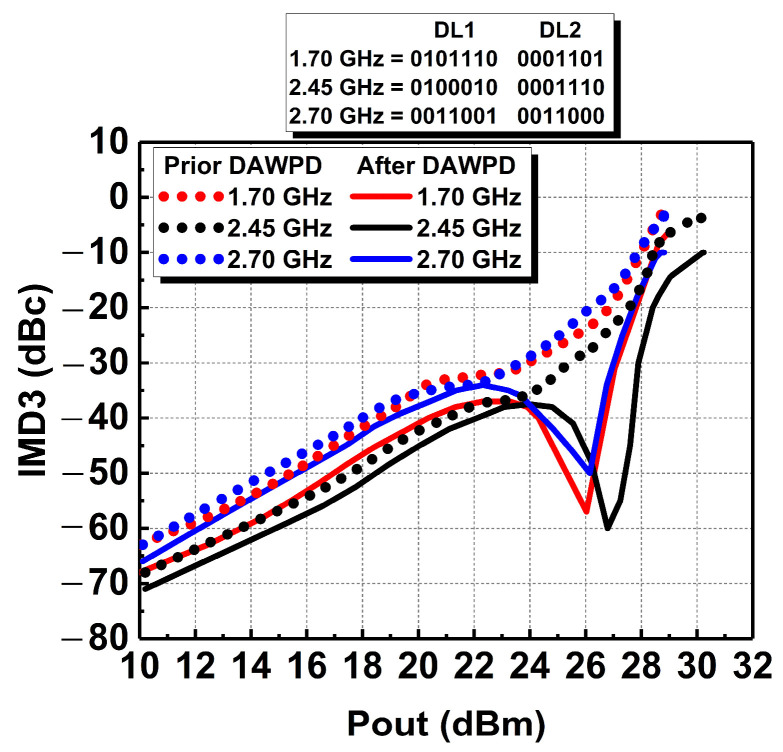
Simulated IMD3 performance prior and after DAWPD at different frequencies.

**Figure 13 micromachines-14-00530-f013:**
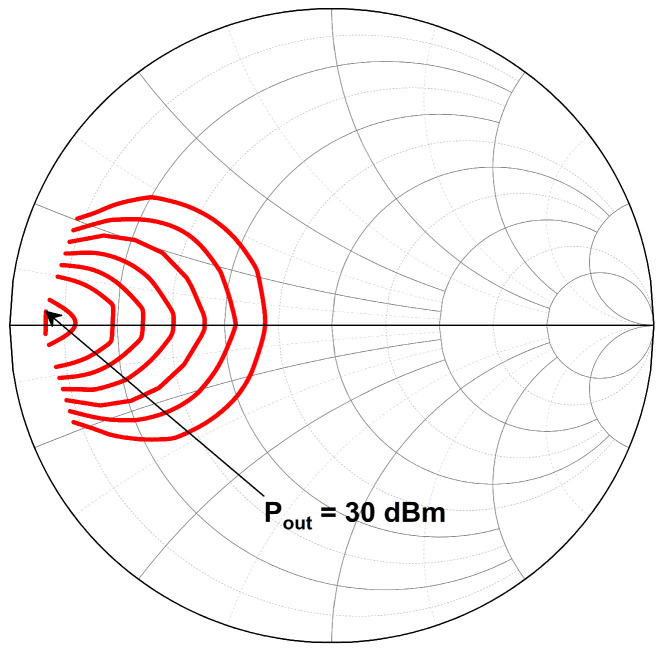
Load-pull analysis of the PA at 2.45 GHz. The load-pull contour is plotted in 1 dB step.

**Figure 14 micromachines-14-00530-f014:**
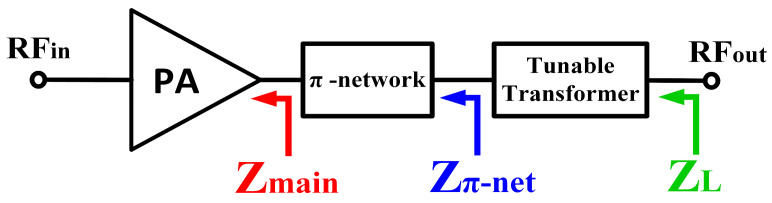
Impedance locations at the output of the DAWPD-PA.

**Figure 15 micromachines-14-00530-f015:**
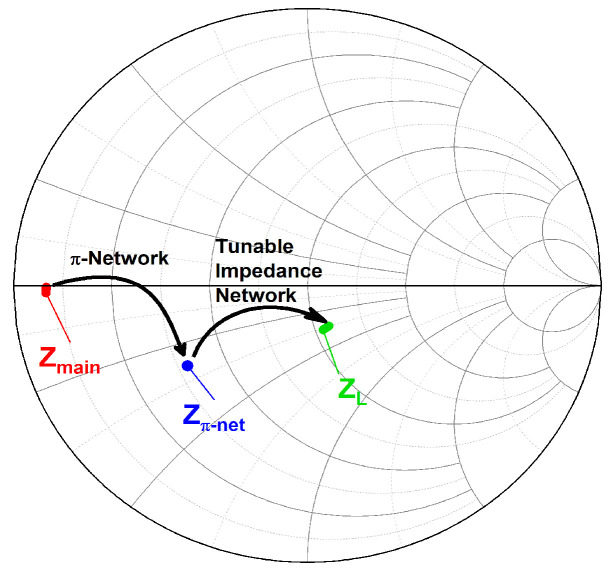
Locations of the output impedances at the drain of the main PA, π network, and tunable impedance network at 2.45 GHz.

**Figure 16 micromachines-14-00530-f016:**
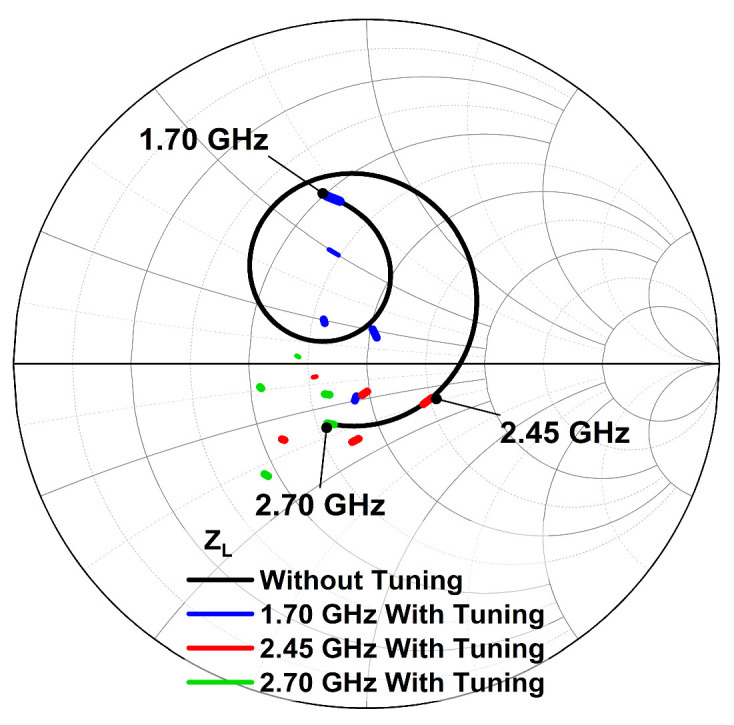
Impedance locations of Z_L_ without and with tuning.

**Figure 17 micromachines-14-00530-f017:**
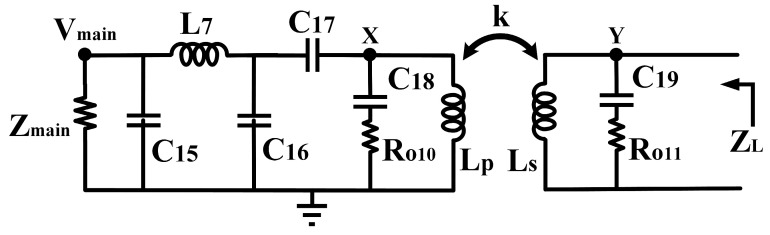
Equivalent circuit of the tunable-output matching network.

**Figure 18 micromachines-14-00530-f018:**
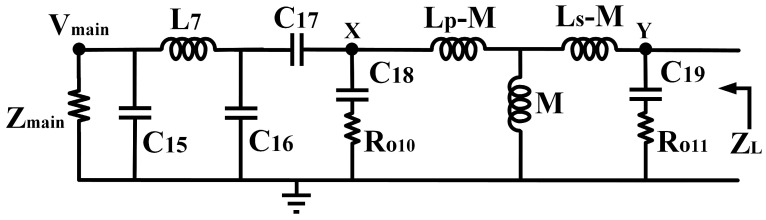
Equivalent circuit of the tunable-output matching network with T-model of the transformer.

**Figure 19 micromachines-14-00530-f019:**
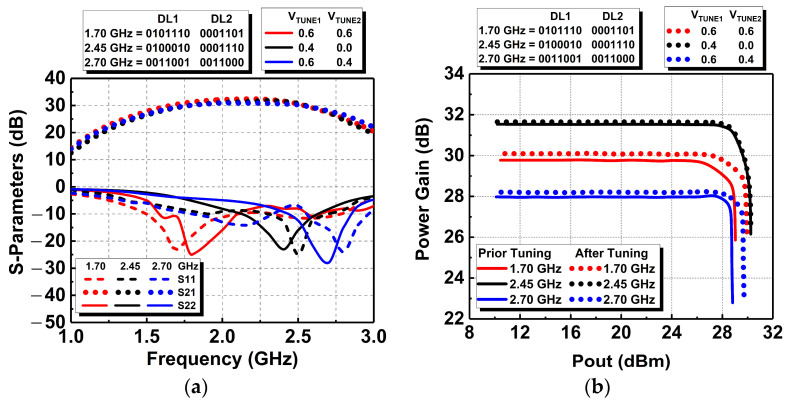
Simulated performances of the DAWPD-PA: (**a**) S parameters, (**b**) power gain and (**c**) PAE prior and after tuning.

**Figure 20 micromachines-14-00530-f020:**
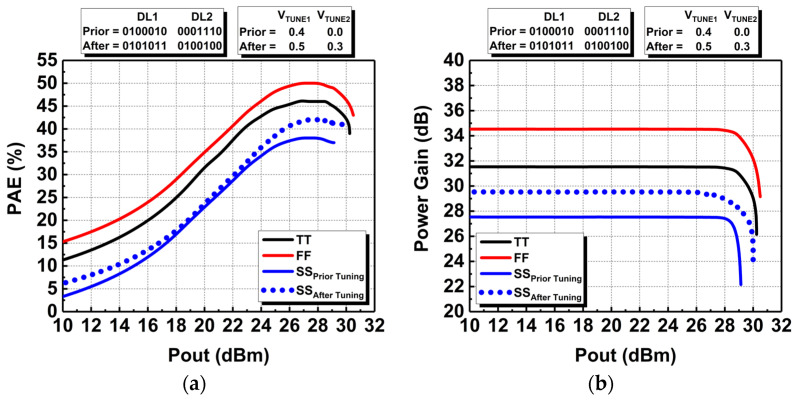
Simulated performances of the DAWPD-PA at different corners: (**a**) PAE and (**b**) power gain prior to and after tuning.

**Figure 21 micromachines-14-00530-f021:**
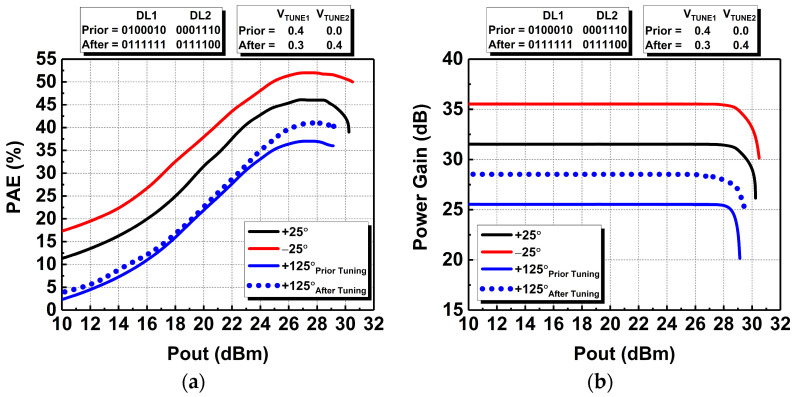
Simulated performances of the DAWPD-PA at different temperatures: (**a**) PAE and (**b**) power gain prior to and after tuning.

**Figure 22 micromachines-14-00530-f022:**
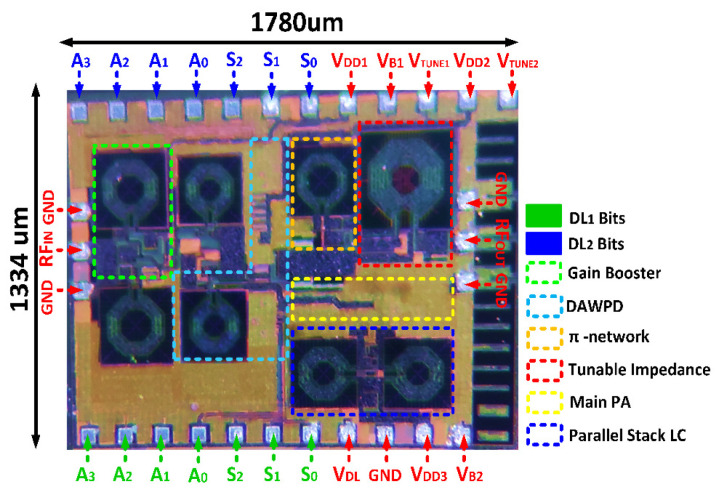
Chip micrograph of the DAWPD-PA.

**Figure 23 micromachines-14-00530-f023:**
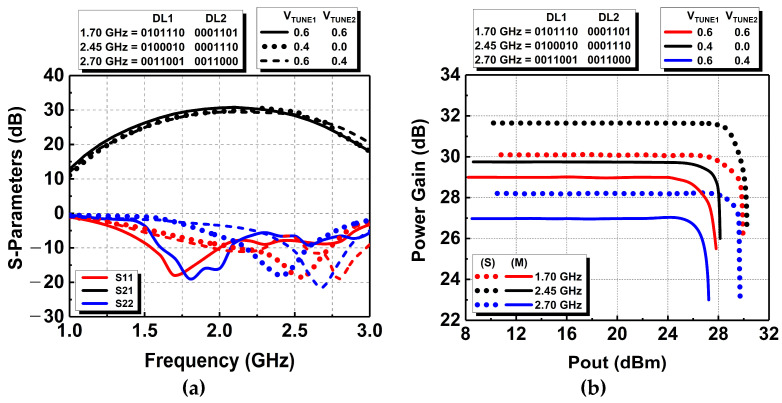
Measured (**a**) S parameters and (**b**) power-gain performances.

**Figure 24 micromachines-14-00530-f024:**
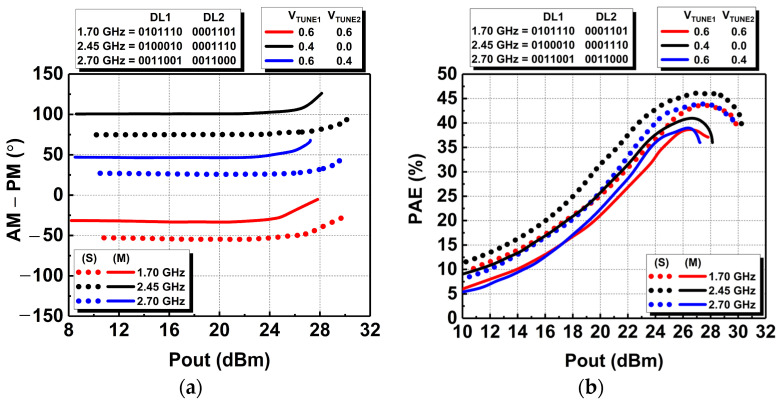
Measured (**a**) phase and (**b**) PAE performances across frequency.

**Figure 25 micromachines-14-00530-f025:**
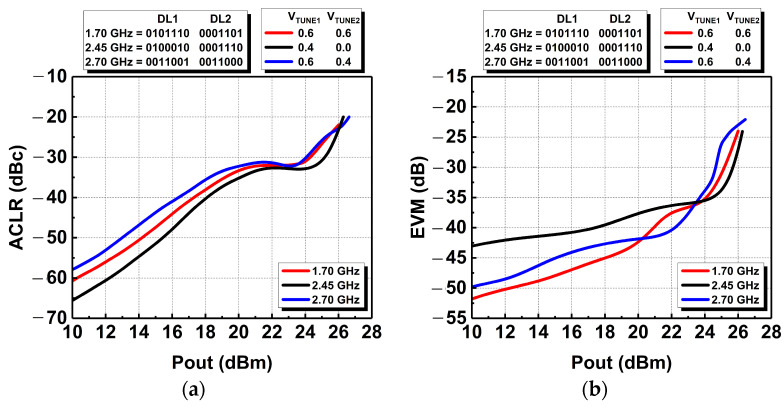
Measured (**a**) ACLR and (**b**) EVM performances across frequencies.

**Figure 26 micromachines-14-00530-f026:**
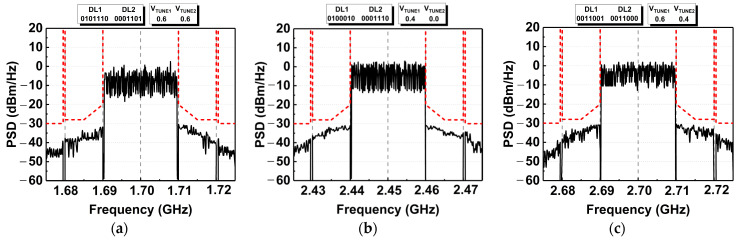
Measured PSD of the DAWPD-PA across the frequencies (**a**) 1.7 GHz, (**b**) 2.45 GHz and (**c**) 2.7 GHz.

**Figure 27 micromachines-14-00530-f027:**
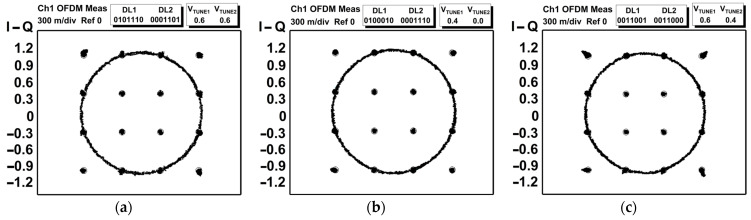
Measured EVM of the DAWPD-PA across the frequencies (**a**) 1.7 GHz, (**b**) 2.45 GHz and (**c**) 2.7 GHz.

**Figure 28 micromachines-14-00530-f028:**
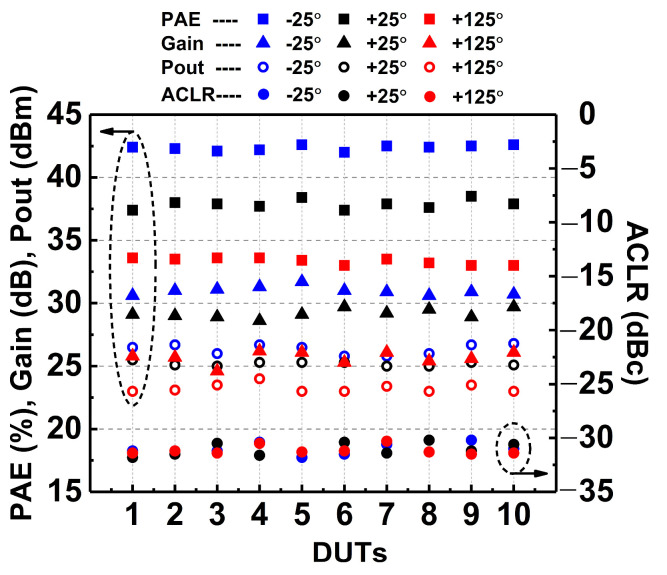
Measured linear PAE, linear Pout, gain, and ACLR performances at 2.45 GHz among 10 samples.

**Table 1 micromachines-14-00530-t001:** Comparison of the CMOS DAWPD-PA with recent works.

Ref.	[10]	[11]	[12]	[29]	[30]	[31]	[32]	TW
Tech. (nm)	40 CMOS	180 CMOS	180 SOI CMOS	65 CMOS	28 CMOS	65 CMOS	40 CMOS	130 CMOS
V_DD_ (V)	1.5	5.0	2.5	2.4	2.2	3.6	2.2	3.3
Freq. (GHz)	1.7–2.1	4.9–5.8	1.9–2.7	1.4–2.5	2.0–2.7	1.65–2.2	0.699–0.915	1.7–2.7
Bandwidth (GHz)	0.4	0.9	0.8	0.9	0.7	0.55	0.216	1.0
Max. Pout (dBm)	26.0–28.2	29.0–31.0	27.5–28.5	20.0–21.7	27.8–28.8	18.7–30.0	25.0–27.1	27.0–28.0
Lin. Pout (dBm)	21.0–23.4	19.8	21.5–22.4	14.0–15.0	23.4	22.8	22.6	24.0–25.1
Max. PAE (%)	25.5–34.0	15.0–25.0	36.0–46.8	25.0–38.1	24.0–30.8	42.4–45.9 *	23.0–33.3	38.8–41.3
Lin. PAE (%)	16.0–23.4	5	18.5–21.7	16.0–24.0	23.2	31.4 *	23.1	34.5–38.8
Gain (dB)	-	25–31	10–11	-	29	-	-	27–29
Channel Bandwidth	20 MHz LTE	20 MHz WLAN	20 MHz LTE	20 MHz LTE	20 MHz WLAN	5 MHz OFDM	1.4 MHz CAT-M1	20 MHz LTE

* drain efficiency.

## Data Availability

Not applicable.

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
