# Peer review of "A Wide-Bandwidth PVT-Reconfigurable CMOS Power Amplifier with an Integrated Tunable-Output Impedance Matching Network"

_micromachines, 2023, doi:10.3390/mi14030530_

Round 1

Reviewer 1 Report

The paper is interesting and well written. There is only one small suggestion for the authors that the DL maybe the digital voltage controller rather than the linearizer  I think. Because the linearization is implemented by the Q4 and Q6. 

Reviewer 2 Report

The paper is very interesting and quite well written. Unfortunately the novelty is not highlighted. There is already the paper  with similar results published and not cited. The already published paper has even same graphics and similar results. Incremental manuscript should bring at least 2/3 of novel results.

Author Response

Please see the latest attachment. The references are revised according to reviewers comments.

Reviewer 3 Report

A very well written article depicting the PA design in all the stages and very well documented in terms of analytical explanation, simulation results and experimental results. Also, a comparison with the state-of-the-art is provided. I have just the couple of comments below that the authors might consider in order to further improve the article.

The first is merely a graphical one: it would be useful if the response of the matching network at different tuning voltage could be shown not only on the Smith Chart, but also as rectangular plots (with frequency as x-axis) to better depict any wideband/resonant behavior.

The second concerns the fact that this type of reconfigurable PA platform might benefit from optimization-based local (or global) tuning/adaptation in order to further improve the performance and account for operating regime and conditions, process variations, etc. For example, in "Mengozzi et al. Automatic Optimization of Input Split and Bias Voltage in Digitally Controlled Dual-Input Doherty RF PAs. Energies 2022, 15, 4892", optimization is applied to the input splitting and bias in order to improve a multi-stage Doherty. In this work, the predistorter and the matching network provide tunable parameters. Could the authors comment?

Again, congratulations for the very nice and detailed work.

Round 2

Reviewer 2 Report

The authors added new figures, which could be optimized to save the space. The authors also improved a reference list. Nevertheless, the main published paper which is actually repetition of this work including conclusions is still not referred. I find this cheating, since in the published paper there are lots of same plots and even conclusions are nearly the same. So there is a lack of novelty. I would be happy if the authors highlight the novelty. This would make the paper more readable. Conclusions of the published paper: IEEE TRANSACTIONS ON CIRCUITS AND SYSTEMS—II: EXPRESS BRIEFS, VOL. 68, NO. 11, NOVEMBER 2021 are: This brief reported a DAAPD-based linearization technique used in a CMOS PA. The silicon chip advances the state-of-the-art linearity close to 1-dB compression point (3-dB back-off) for 1-GHz bandwidth from 1.7 to 2.7 GHz with a linear PAE above 35%. We confirmed it with a very competitive ACLR performance under a 20-MHz LTE channel bandwidth where the DAAPD-PA satisfied the specification of −33 dBc. The measured linear output power ranges from 24 to 25 dBm considering a 3-dB back-off from a saturated output power of 27 to 28 dBm. The DAAPD’s digital tuning permitted the reconfiguration of the PA for multiband applications insensitive to PVT variation. Conclusions of this paper: The 130 nm wideband CMOS DAWPD-PA with tunable output impedance matching 705 network has delivered an operating bandwidth from 1.7 to 2.7 GHz. The DAWPD with a digital implementation and active load offers a tunable wideband linearization across the aforementioned frequency. It also utilizes a tunable impedance matching network realized via a transformer and tuner circuits to reconfigure its performances. The maximum output power attained by the wideband DAWPD-PA is 27 to 28 dBm with a peak PAE of 38.8 to 41.3%. A linear output power of 24 to 25.1 dBm with linear PAE of 34.5 to 38.8% is achieved when tested with a 20 MHz LTE modulated signal. The proposed configurability of the DAWPD-PA also makes it resilient to PVT variations and enhances its reliability. Its performances can be maintained via the design's tuning property, which is proven when 10 DUTs were tested across different corners and temperatures. The tuning of the digital bits states of the DAWPD and the tuning of the tunable output impedance matching network realize the reconfigurable wideband CMOS PA. => not much difference. My conclusion is that the paper repetition in another journal is not a good idea and might create problems from IEEE. If they want to publish this manuscript, the novelty should be highlighted and the already published plots should be cited. I see some added value, approximately by 30%, compared to the published paper. This is not enough. It is not clear why the authors "forgot" their own paper which is very close to this one. Novelty should be highlighted and should contain more than 50% of the already published paper. All replotted graphs or similar ones should be cited with permission from IEEE to avoid problems from IEEE. To my opinion the manuscript could be accepted with major revision.

Author Response

Please see the attachment. The corrections done are highlighted in yellow in a separate manuscript in order to ease the reviewing process for the reviewer. Thank you very much.

Round 3

Reviewer 2 Report

 Authors have improved the paper noticeably by adding more than 30% of research material .  The nodes in the circuits in Fig.: 6, 9, 18 are missing.

Author Response

Thank you vey much for the comments to greatly improve our papers. Please see the attachment for the response.
